



**Sources of non-fossil fuel emissions in carbonaceous aerosols during early winter in Chinese**
**cities**
Di Liu[1], Jun Li[1*], Zhineng Cheng[1], Guangcai Zhong[1], Sanyuan Zhu[1], Ping Ding[2], Chengde Shen[2],
Gan Zhang[1]
[1]State Key Laboratory of Organic Geochemistry, Guangzhou Institute of Geochemistry, Chinese
Academy of Sciences, Guangzhou, 510640, China
[2]State Key Laboratory of Isotope Geochemistry, Guangzhou Institute of Geochemistry, Chinese
Academy of Sciences, Guangzhou, 510640, China
*To whom correspondence may be addressed:
Dr. Jun Li; Email: junli@gig.ac.cn; Tel: +86-20-85291508; Fax: +86-20-85290706
**Abstract**
China experiences frequent and severe haze outbreaks from the beginning of winter.
Carbonaceous aerosols are regarded as an essential factor in controlling the formation and
evolution of haze episodes. To elucidate the carbon sources of air pollution, source apportionment
was conducted using radiocarbon ($^{14}$C) and unique molecular organic tracers. Daily 24-hour $PM_{2.5}$
samples were collected continuously from October 2013 to November 2013 in 10 Chinese cities.
The $^{14}$C results indicated that non-fossil fuel (NF) emissions were predominant in total carbon (TC;
average = 65 ± 7%). Approximately half of the EC was derived primarily from biomass burning
(BB) (average = 46 ± 11%), while over half of the OC fraction comprised NF (average = 68 ± 7%).
On average, the largest contributor to TC was NF-derived secondary OC ($SOC_{nf}$), which
accounted for 46 ± 7% of TC, followed by SOC derived from fossil fuels (FF) ($SOC_f$; 16 ± 3%),
BB-derived primary OC ($POC_{bb}$; 13 ± 5%), POC derived from FF ($POC_f$; 12 ± 3%), EC derived
from FF ($EC_f$; 7 ± 2%) and EC derived from BB ($EC_{bb}$; 6 ± 2%). The regional background
carbonaceous aerosol composition was characterized by NF sources; POCs played a major role in





northern China, while SOCs contributed more in other regions. However, during haze episodes,
there were no dramatic changes in the carbon source or composition in the cities under study, but
the contribution of POC from both FF and NF increased significantly.

**1.  Introduction**
Recently, a wide range of fine particle (PM$_{2.5}$) pollution has affected northern, central and
southern China, particularly on haze days, which has had significant effects on air quality,
atmospheric visibility and public health, and caused extensive public and scientific concern (Liu et
al., 2013a;Wang et al., 2014). Haze events in Chinese urban areas, especially in megacities, have
become a common phenomenon, appearing in every season, because of large and intensive
pollutant emissions and unfavorable meteorological conditions (He et al., 2014;Liu et al., 2013b).
Generally, heavy and serious haze pollution outbreaks start at the beginning of winter.
Carbonaceous aerosols are the dominant component of PM$_{2.5}$ (~20–80%) (Rogge et al.,
1993;He et al., 2004;Dan et al., 2004;Kanakidou et al., 2005) and are regarded as essential for
controlling the formation and evolution of haze episodes. Relatively high concentrations of
carbonaceous aerosols have been observed during typical haze days in northern, southern and
central China (Zhao et al., 2013;Deng et al., 2008;Zhang et al., 2014a). Generally, carbonaceous
aerosols (total carbon, TC) can be divided into elemental carbon (EC) and organic carbon (OC)
according to their different physical and chemical properties (Krivácsy et al., 2001;Kleefeld et al.,
2002). EC is formed either from biomass burning (BB; e.g., wood fires, heating) or fossil fuels
(FF; e.g., vehicle or industry emissions), and can be used as a tracer for primary
combustion-generated OC because primary OC and EC are mostly emitted from the same sources
(Turpin and Huntzicker, 1995;Strader et al., 1999). OC can be directly derived from primary



emissions (primary OC; POC), or formed through oxidation of reactive organic gases followed by
gas-to-particle conversion in the atmosphere (secondary OC; SOC). Moreover, further
subcategories of OC exist, including water-soluble organic carbon (WSOC) and water-insoluble
organic carbon (WINSOC), which are distinguished on the basis of water-solubility; these may be
essential for assessing the different sources of OC emissions during haze episodes, since WSOC is
a proxy for SOC and BB OC, while WINSOC better represents POC (Weber et al.,
2007b;Docherty et al., 2008;Mayol-Bracero et al., 2002;Weber et al., 2007a); (Huang et al., 2014).
Several methods have been introduced to identify and quantify OC emission sources, such as
the use of organic molecular tracers (Simoneit et al., 1999); however, their reliability is limited by
their low atmospheric lifetimes, in turn due to chemical reactivity and highly variable emission
factors (Fine et al., 2001, 2002, 2004;Gao et al., 2003;Hedberg et al., 2006;Robinson et al., 2006).
Recently, radiocarbon ($^{14}$C) analysis has been used as a powerful tool for facilitating the direct
differentiation of non-fossil fuel (NF) carbon sources from fossil fuel (FF) sources, because $^{14}$C is
completely absent from FF carbon (e.g., diesel and gasoline exhaust, coal combustion), whereas
NF carbon (e.g., biomass burning, cooking and biogenic emissions) shows a high contemporary
$^{14}$C level (Szidat et al., 2009) Hence, $^{14}$C measurements can provide information about the
contributions of FF, BB and biogenic emissions to carbonaceous aerosols. Numerous studies have
been performed on the regional background of carbonaceous aerosols at urban sites; for example,
contemporary carbon was the dominant pollutant in carbonaceous aerosols at a background site;
while a significant difference was found among seasons at urban sites (Yang et al., 2005;Chen et
al., 2013;Liu et al., 2013a;Zhang et al., 2014b;Liu et al., 2014a). A combination of $^{14}$C analysis
and organic tracer determination allows for more detailed source apportionment of carbonaceous



aerosols (Gelencsér et al., 2007;Ding et al., 2008;Lee et al., 2010;Yttri et al., 2011).
In this study, sampling was conducted in 10 typical Chinese cities during early winter, i.e., at
the beginning of the period of widespread hazes. Carbonaceous aerosols, including different
carbon fractions such as WSOC, WINSOC and EC, along with water-soluble inorganic ions ($F^-$,
$Cl^-$, $SO_4^{2-}$, $NO_3^-$, $NH_4^-$, $Na^+$, $K^+$, $Ca^{2+}$ and $Mg^{2+}$) and anhydrosugars (levoglucosan, galactosan and
mannosan), were analyzed in $PM_{2.5}$ samples. Source apportionment of carbonaceous aerosols was
performed using [14]C and organic tracers.
**2. Materials and Methods**
**2.1 Aerosol sampling**
Daily 24-hour $PM_{2.5}$ samples were collected continuously on the rooftops of institutes in 10
Chinese cities (Figure 1) from October 2013 to November 2013. In total, 292 aerosol samples,
including 10 field blanks, were collected on pre-heated (450ºC for 5 h) quartz fiber filters (8 × 10
inches; Whatman, UK) using a high volume sampler with a flow rate of 0.3 $m^3$ $min^{-1}$. The filters
were then wrapped in aluminum foil, packed into air-tight plastic bags, and stored at -20°C in a
refrigerator until analysis. $PM_{2.5}$ mass concentrations were determined gravimetrically by state
regulatory agencies. Details of the sampling information and meteorological parameters used
during sampling are shown in the Supporting Information (SI).
**2.2 Chemical analysis**
OC and EC were obtained with an off-line carbon analyzer (Sunset Laboratory, Inc., USA) using
the thermo-optical transmittance method (NIOSH 870). Water-soluble inorganic ions ($Na^+$, $Cl^-$,
$Ca^{2+}$, $Mg^{2+}$, $K^+$, $NH_4^+$, $SO_4^{2-}$ and $NO_3^-$) were analyzed with an ion chromatographer (83
Basic IC Plus, Metrohm, Switzerland). Anhydrosugars (levoglucosan, galactosan and mannosan)





were analyzed by gas chromatography-mass spectroscopy (GC-MS) (7890-5975; Agilent) using a
capillary column (DB-5MS; 30m, 0.25 mm, 0.25μm). Analysis methods related to OC and EC,
water-soluble inorganic ions (Wang et al., 2012) and anhydrosugars (Liu et al., 2014a;Liu et al.,
2014b) were presented elsewhere and a detailed analytical procedure and method are available in
the SI.
**2.3 Separation of carbon species**
A punched section of filtrate was cut and sandwiched in a filtration unit, then extracted with 100
mL ultra-pure water (18.2 MΩ). WSOC species were quantified using a total organic carbon (TOC)
analyzer (TOC-VCPH; Shimadzu, Japan). The punched filtrate was dried in a desiccator, wrapped
in aluminum foil and then stored in a refrigerator. WINSOC and EC were obtained from the
water-filtered sample with an off-line carbon analyzer (Sunset Laboratory, Inc.) using the
thermo-optical transmittance method (NIOSH 870).
**2.4 Radiocarbon measurements**
Isolation procedures for the $^{14}$C measurements of WSOC, WINSOC and EC have been
described previously (Liu et al., 2016a; Liu et al., 2013a). Two filters, based on the $PM_{2.5}$
concentrations at each site, were used for $^{14}$C determination of WSOC, WINSOC and EC, to
distinguish between FF and NF emissions. To obtain the WSOC, WINSOC and EC fractions from
a single punch filter, a circular section of the punch filter was clamped in place between a filter
support and a funnel and then 60 ml ultra-pure water was slowly passed through the punch filter
without a pump, allowing the WSOC to be extracted delicately. WSOC was quantified as the total
dissolved organic carbon in solution using a total organic carbon (TOC) analyzer (Shimadzu
TOC_VCPH, Japan) following the nonpurgeable organic carbon protocol. WSOC solution was



freeze-dried to dryness at -40 °C. The WSOC residue was re-dissolved with ~500 µl of ultra-pure
water and then transferred to a pre-combusted quartz tube, which was then placed in the freeze
dryer. After that, the quartz tube was combusted at 850 °C. The remaining carbon on the filter was
identified as WINSOC or EC by an OC/EC analyzer (Sunset, U.S.). After WSOC pretreatment and
freeze-dried, OC is oxidized to $CO_2$ under a stream of pre-cleaned oxygen pure analytical grade
$O_2$ (99.999%, 30 ml min$^{-1}$) during the pre-combustion step at 340 °C for 15 min. Before the OC is
oxidized, the sample is first positioned in the 650 °C oven for about 45 s flash heating. This flash
heating has the advantage of minimizing pre-combustion charring, since it reduces pyrolysis of
OC. After the OC separation, the filters were removed from the system, placed into a muffle
furnace at 375°C, and combusted for 4 h. The filters were then quickly introduced back into the
system and oxidized under a stream of pure oxygen at 650°C for 10 min to analyze the EC
fraction. Finally, the corresponding evolved $CO_2$ (WSOC, WINSOC, and EC) was cryo-trapped,
quantified manometrically, sealed in a quartz tube and reduced to graphite at 600 °C using zinc
with an iron (200 mg, Alfa Aesar, 1.5-3 mm, 99.99%) catalyst for accelerator mass spectrometry
(AMS) target preparation. Approximately 200 µg of carbon was prepared for each carbon fraction.
All $^{14}$C values were reported as the fraction of modern carbon ($f_m$) after correcting for
fractionation with $\delta^{13}$C. The degree of uncertainty in the $^{14}$C measurements was in the range of
0.2–0.6%. In this study, $f_m$ was converted to the fraction of contemporary carbon ($f_c$), to eliminate
the effects of nuclear bomb tests through application of conversion factors of 1.10 ± 0.05 for EC
and 1.06 ± 0.05 for 2013 OC data. Here, the $f_m$ values of OC (OC = WSOC + WINSOC) and TC
(TC = WSOC + WINSOC + EC) were calculated by isotopic mass balance. The concentration in
the field blank was negligible (0.37 ± 0.05 µg cm$^{-2}$; less than 5% carbon) and no field blank





subtraction was made for $^{14}$C determination. The system blank F$^{14}$C was 0.0036(SD=0.0001),
which translated to a $^{14}$C age of around 45,000 years BP.
**3. Results and Discussion**
**3.1 PM$_{2.5}$, OC and EC concentrations and spatial distribution**
PM$_{2.5}$ levels ranged from 21.9 to 482 µg m$^{-3}$, with an average level of 178 ± 103 µg m$^{-3}$. A total of
98% and 81% of PM$_{2.5}$ exceeded the First Grade National Standard (35 µg m$^{-3}$) and Second Grade
National Standard (75 µg m$^{-3}$) of China, respectively, indicating relatively poor air quality during
sampling days. The OC and EC levels ranged from 0.99 to 75.9 µg m$^{-3}$ (average = 22.8 ± 15.3 µg
m$^{-3}$) and 0.07 to 19.3 µg m$^{-3}$ (average = 3.66 ± 3.28 µg m$^{-3}$), respectively; thus, OC and EC were
major components of PM$_{2.5}$, accounting for 13 ± 8% and 2 ± 1% of PM$_{2.5}$, respectively. The OC
and EC levels in this study were generally higher than those recorded previously in more
developed cities (e.g., New York, Los Angeles, Erfurt, Kosan) (Kam et al., 2012;Kim et al.,
2000;Gnauk et al., 2005;Rattigan et al., 2010), indicating severe carbonaceous pollution and
emphasizing the importance of restricting carbonaceous aerosols in China.
Northern China has high PM$_{2.5}$ concentrations. As shown in Table 1, the average PM$_{2.5}$
concentrations in Beijing (190 ± 79 µg m$^{-3}$), Xinxiang (245 ± 65 µg m$^{-3}$), Taiyuan (285 ± 84 µg
m$^{-3}$) and Lanzhou (212 ± 112 µg m$^{-3}$) were significantly higher than those in central and southern
China (from 85 µg m$^{-3}$ in Guangzhou to 123 µg m$^{-3}$ in Wuhan). Shanghai, in the eastern coastal
region, had the lowest average PM$_{2.5}$ concentration (67 ± 43 µg m$^{-3}$). The ratio of total organic
matter (TOM; 1.6 × OC + EC) to total fine particle mass ranged from 17.4% to 32.6%, except in
Guiyang. Cities in central and southern China, such as Chengdu, Wuhan, Nanjing, and Guangzhou,
had a higher ratio of TOM to PM$_{2.5}$ than other cities. Moreover, the OC/EC ratios in those cities



were also higher, with values ranging between 8.1 and 12. The spatial distribution pattern closely
reflected energy consumption and regional climate differences. For example, there were more
particle emissions from heating in northern China, and more secondary organic aerosols in
southern and central China. In particular, Guiyang, which is a developing city located on the
Western plateau, had a high level of $PM_{2.5}$ ($227 \pm 77$ μg m$^{-3}$), comparable to that in northern China,
but also had the lowest levels of OC and EC. Moreover, the TOM to $PM_{2.5}$ ratio was only about
6.0%. This indicates that there are different chemical sources in this developing city compared to
megacities in China.
**3.2 Radiocarbon results: fraction of modern carbon ($f_m$)**
Table 2 shows the proportion (%) of NF sources in various carbon fractions. Overall, NF
emissions represented a more significant proportion of the TC (average = $65 \pm 7\%$; range:
50–79%), at all sites, than FF sources, which underscores the importance of NF sources to
carbonaceous aerosols during early winter in China.
EC is only formed by primary emissions, which are inert in ambient air and originate either
from BB or FF combustion. In this study, about half of the EC was derived from BB in the 10
urban cities (average $46 \pm 11\%$; range: 24–71%), which represents a slightly higher proportion
than that for the same cities in winter and spring, but is similar to previous studies performed in
cities in other countries (Szidat et al., 2009;Bernardoni et al., 2013;Liu et al., 2016b). However,
this result differs from those obtained in remote regions dominated by BB (Barrett et al.,
2015;Zhang et al., 2014b). A larger contribution of BB to EC was found in central and western
China (i.e., Beijing, Lanzhou, Chengdu and Guiyang) (49~63%), where Guiyang had the largest
proportion of BB in EC ($63 \pm 12\%$), followed by Beijing ($50 \pm 2.0\%$), Chengdu ($50 \pm 1.8\%$),





Wuhan (48 ± 10%) and Nanjing (47 ± 5%); this shows that there are large amounts of BB
emissions (e.g., from biofuel burning and outdoor fires) in western and central China during early
winter. This phenomenon was also found in central China during the severe haze episode that
occurred over China in January 2013, which suggests that these massive BB emissions were
generated indoors (i.e., from domestic heating and cooking) and thus could not be detected by
MODIS [*Liu et al.*, 2016b]. Guangzhou had the lowest proportion of BB in EC (32 ± 12%),
suggesting that FF emissions (coal combustion and vehicle emissions) dominated in the Pearl
Delta region. Similar to Guangzhou, Taiyuan and Xinxiang had lower proportions of BB in EC, of
36 ± 11% and 37 ± 1.7%, respectively. High proportions of BB in EC are due to extremely high
levels of BB tracers (levoglucosan). In this study, levoglucosan concentrations were in the range
161 to 672 ng m$^{-3}$ (377 ± 153 ng m$^{-3}$), and were significantly correlated with EC concentrations in
BB (r = 0.708, p=0.000).
Over half of the OC fraction was from NF sources at all sites (range: 54–82%), with an
average NF source contribution of 68 ± 7%. Generally, the $f_m$ spatial distribution of OC is similar
to that of EC, with NF sources contributing more in central China. Here, OC was divided into
WSOC and WINSOC, which has been separated with respect to fossil and NF sources. A large
contribution of NF sources to WINSOC (64 ± 7%) was observed in this study, comparable to
previous studies performed in urban areas of Europe, e.g., Gothenburg (55 ± 8%) and Zurich (70 ±
7%) (Szidat et al., 2009;Zhang et al., 2013). Moreover, the $f_m$ values for WSOC (70 ± 8%) were
slightly higher than those for WINSOC, which showed values comparable to those observed in
European and American cities (~70−85%) (Weber et al., 2007a;Szidat et al., 2009;Zhang et al.,
2013). A higher $f_m$ value indicated that, for WSOC, the contribution of NF emission sources was





greater. WSOC is regarded as a mixture of SOC and BB-derived POC, whereas WINSOC is
mainly composed of POC from FF combustion, BB and biogenic sources. In this study, the ratio
of WSOC to OC increased significantly with an increase in the proportion of NF sources in OC (r
= 0.531,p=0.016); this implies that POC from BB is more water-soluble, or that more NF-derived
VOCs were involved in SOC formation.
**3.3 Source apportionment of different carbon fractions**
A source apportionment model for carbonaceous aerosols, including primary and secondary
sources, was applied in this study using measured carbon fractions, anhydrosugars, and $^{14}$C
isotopic signals. Detailed information on this model has been provided previously (Liu et al.,
2014a;Liu et al., 2016b).
Briefly, EC from FF combustion ($EC_f$) and BB-derived EC ($EC_{bb}$) can be estimated using the
following respective equations:
$$EC_f = EC \times (1\text{-}f_c) \qquad\qquad [1]$$
$$EC_{bb} = EC \times f_c \qquad\qquad [2]$$
Similar to EC, OC can be divided into FF OC ($OC_f$) and NF OC ($OC_{nf}$) based on $^{14}$C
concentrations. $OC_{nf}$ consists of BB-derived primary OC ($POC_{bb}$), NF-derived SOC ($SOC_{nf}$) and
biological primary carbon (BPC), such as spore and plant debris. BPC particles exist mainly in
coarse fractions (> 2.5 μm) and only account for ~1% of OC in $PM_{2.5}$ [Guo et al., 2012]. Thus, this
carbon fraction was ignored in the present study. $POC_{bb}$ can be semi-quantitatively estimated from
Lev concentrations, due to its unique characteristic of originating from BB, as follows:
$$POC_{bb} = Lev \times (OC/Lev)_{bb} \qquad\qquad [3]$$
According to the levoglucosan/mannosan (Lev/Man; 17.4 ± 5.9) and mannosan/galactosan





(Man/Gal; 2.1 ± 0.3) ratios obtained in this study, 7.7 6 ± 1.47 was adopted as the $(OC/Lev)_{bb}$
value [Liu et al., 2014].
Thus, the $SOC_{nf}$ fraction can be estimated through subtraction:
$$SOC_{nf} = OC_{nf} - POC_{bb} \qquad [4]$$
FF-derived POC and SOC can be estimated by the following respective equations:
$$POC_f = WINSOC \times (1-f_c) \qquad [5]$$
$$SOC_f = WSOC \times (1-f_c) \qquad [6]$$
Figure 2 shows the proportions of different carbon fractions, including $EC_f$, $EC_{bb}$, $POC_{bb}$, $POC_f$,
$SOC_{nf}$ and $SOC_f$, in total carbon (TC) for the 10 urban cites during the sampling period. On
average, the largest contributor to TC was $SOC_{nf}$, accounting for 46 ± 7% of TC, followed by
$SOC_f$ (16 ± 3%), $POC_{bb}$ (13 ± 5%), $POC_f$ (12 ± 3%), $EC_f$ (7 ± 2%) and $EC_{bb}$ (6 ± 2%). The
proportion of primary sources ($POC_{nf} + POC_f + EC_{nf} + EC_f$) (average = 38 ± 9%; range: 25–56%)
was lower than that of secondary sources ($SOC_{nf} + SOC_f$) (average = 62 ± 9%; range: 35–83%),
which underlines the importance of SOC in carbonaceous pollution.
It should be noted that the model uncertainties in these contributions depended mainly on
correction factors, such as the $(POC/Lev)_{bb}$ emission ratios for wood burning, and on conversion
factors used for determining the $f_c$ in $^{14}C$ analysis. Typical relative uncertainties were recently
estimated, using a similar modelling approach, at 20–25 % for $SOC_{nf}$, $SOC_f$, $POC_{bb}$, and $POC_f$,
and ~13% for $EC_f$, and $EC_{bb}$ (Zhang et al., 2015).
POC and EC aerosols are independent from atmospheric gas reaction conditions and thus
directly reflect the characteristics of local emission sources. The total proportions of $EC_f$ and $POC_f$
ranged from 10–38%, with an average of 19 ± 9% for all sites. The total proportions of $EC_f$ and



POC$_f$ in northern and southern China were greater than in western central and eastern coastal
China, indicating a higher impact of FF on local air pollution in both regions. The ratios of POC$_f$
to EC$_f$ (0.66–3.32) were comparable to those derived directly from industrial coal combustion
(2.7–6.1) (Zhang et al., 2008) and traffic exhausts fumes (0.5–1.3) (Zhou et al., 2014;He et al.,
2008), indicating that industrial coal combustion and traffic exhaust fumes were the major primary
sources at all sites. Beijing (2.6) and Xinxiang (3.3) were mainly dominated by coal combustion
emissions. The total proportions of EC$_{bb}$ and POC$_{bb}$ ranged from 12–36%, with an average of 19 ±
8%. West central cities, such as Lanzhou, Chengdu, Guiyang, Nanjing and Wuhan, had large
proportions of EC$_{bb}$ and POC$_{bb}$ (average = 23 ± 7%; range: 14–36%), which confirms the greater
impact of BB on local air pollution in West central China; this should be considered when setting
future limits for polluting corporations.
Total SOC in OC ranged from 42–84% (average = 72 ± 10%) among the sites tested in this study,
which is similar to recent studies, conducted in the haze period in China of January 2013, which
used high-resolution aerosol mass spectrometry; i.e., 41–59% [Sun et al., 2014] and 44–71%
[Huang et al., 2014] obtained from online and offline measurements, respectively. There was no
significant difference in the SOC/OC ratio among the different regions in China studied herein,
except for Guiyang, which had a somewhat lower SOC/OC ratio. Moreover, SOC was comprised
predominantly of NF sources at all sites (67–89%), except at Guiyang with values of 42-53%,
which are similar to areas in developed countries with good air quality, such as Puy de Dôme,
France (86–88%) and Schauinsland, Germany (84–93%) [Gelencsér et al., 2007]. However, our
values were higher than those of previous studies conducted in China during other winter and
spring seasons, indicating the importance of NF to SOC in China during early winter.



### 3.4 Comparison of chemicals between samples by PM$_{2.5}$ concentration

Two samples, one each with a low and high PM$_{2.5}$ concentration, were obtained from all 10 study

sites (Figure S1) for [14]C and inorganic ions analysis, to investigate the composition of

carbonaceous aerosols and evaluate the importance of FF and NF carbon in haze formation across

China in early winter. During sampling, the air masses generally moved in a northwesterly to

northeasterly direction to reach the site. The 5-day back trajectory analysis revealed relatively

lower concentrations of PM$_{2.5}$ when the wind speed was higher, and relatively higher PM$_{2.5}$ levels

when the wind speed was lower and more stable; synoptic conditions apparently promoted the

accumulation of particles (Figure 3).

Theoretically, the aerosol composition at higher wind speeds should reflect regional background

aerosol characteristics. Figure 3 shows the PM$_{2.5}$ chemical compositions of the stage for lower

PM$_{2.5}$ concentration during sampling period. Here, due to the different conversion factors used to

transform WINSOC to WINSOM (1.3), and WSOC to WSOM (2.1), OM calculations were based

on the relative contributions of WSOC and WINSOC to OC. TOM is the sum of EC, WINSOM

and WSOM. Generally, TOM contributions to PM$_{2.5}$ ranged from 21–38%, except in Guiyang

where a value of 8% was observed. Moreover, OM was comprised mainly of NF emissions. In

cities in northern China (Beijing, Xinxiang and Taiyuan), the contribution of WINSOM (both FF

and NF) was greater, indicating that POC played a major role in regional air quality during this

season. Simultaneously, the lower NO$_3^-$/SO$_4^{2-}$ ratios also implied that POC from FFs might be

derived predominantly from coal combustion. The 5-day back trajectory analysis showed that the

air mass came from northern China, including regions such as Inner Mongolia and Hebei province,

where the ambient temperature is always below 10ºC during this season. It is very common for





local rural residents to burn coal or biomass fuel to generate heat for their households. Therefore,
coal and biomass fuel combustion in northern China might be the major contributor to regional
carbonaceous aerosols in northern China during this season. In other cities, WSOM levels in both
FF and NF were much higher than those in WINSOM, showing the importance of SOC across
China. However, $NO_3^-/SO_4^{2-}$ ratios in Shanghai, Nanjing and Wuhan were much higher than in
other areas. The back trajectory results showed that the air mass came from northern China or the
Yangtze River Delta, implying that traffic exhaust emissions in those regions was more important
for carbonaceous aerosol composition.

300       The chemical compositions of the higher $PM_{2.5}$ samples obtained in each city are shown in

Figure 3. There were no dramatic changes in the carbon source or composition in any of the cities;
however, the contribution of EC and WINSOM to both fossil and NF fuels increased significantly,
along with the $NO_3^-/SO_4^{2-}$ ratios, indicating the importance of POC from local regions. The back
trajectory results showed that wind speeds were moderate and stable, and that synoptic conditions
apparently promoted the accumulation of particles derived either from local or regional sources.
**4. Conclusion**

307       $PM_{2.5}$ samples were collected continuously from 10 Chinese urban cities during early winter

2013. $PM_{2.5}$, OC and EC levels were highest in northern China, with maximum concentrations of
482 µg m$^{-3}$, 75.9 µg m$^{-3}$ and 19.3 µg m$^{-3}$, respectively. OC and EC were the major components of
$PM_{2.5}$, accounting for 13 ± 8% and 2 ± 1%, of total $PM_{2.5}$, respectively. The $^{14}C$ results, for the
lower and higher $PM_{2.5}$ concentration sample pairs obtained at each city, indicated that, overall,
NF emissions constituted a significant proportion of TC (average = 65 ± 7%) at all sites, i.e.,
higher than FF sources. Furthermore, about half of the EC was derived primarily from BB



(average = 46 ± 11%), and over half of the OC fraction came from NF sources (average = 68 ±
7%). Source apportionment analysis was done using $^{14}$C and unique molecular organic tracers. On
average, the largest contributor to TC was $SOC_{nf}$, accounting for 46 ± 7% of TC, followed by
$SOC_f$ (16 ± 3%), $POC_{bb}$ (13 ± 5%), $POC_f$ (12 ± 3%), $EC_f$ (7 ± 2%) and $EC_{bb}$ (6 ± 2%). When
relatively lower $PM_{2.5}$ concentrations were observed, OM was dominant in carbonaceous aerosols,
mainly from NF. POC played a major role in regional air quality in the cities in northern China,
while SOC contributed more in cities in other regions of China. There were no dramatic changes
in carbon sources or carbon compositions in the sampled cities during haze days; however, the
contribution of POC from both NF and NF increased significantly in these periods. This indicates
that synoptic conditions promote the accumulation of particles derived either from local or
regional sources.

**Acknowledgements**
This study was supported by the "Strategic Priority Research Program (B)" of the Chinese
Academy of Sciences (Grant No. XDB05040503), the Natural Science Foundation of China
(NSFC; Nos. 41430645, 41473101 and 41503092), and the Guangzhou Science and Technology
Plan Projects (No. 201504010002). All data in this manuscript are freely available on request
through the corresponding author (junli@gig.ac.cn).This is a contribution of GIGCAS.

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





Table 1 The $PM_{2.5}$, OC and EC data used in this study (average ± standard deviation; µg m$^{-3}$)

| Sites | N | $PM_{2.5}$ | OC | EC | OM/$PM_{2.5}$ (%) | OC/EC |
|---|---|---|---|---|---|---|
| Beijing | 31 | 189±79 | 26.5±12.5 | 3.6±1.8 | 24±4.6 | 7.7±1.8 |
| Xinxiang | 31 | 245±65 | 29.3±11.7 | 4.8±2.2 | 21±4.9 | 6.5±1.9 |
| Taiyuan | 31 | 285±84 | 37.3±15.5 | 7.8±2.8 | 23±4.4 | 4.9±1.5 |
| Lanzhou | 31 | 212±112 | 21.4±9.1 | 5.0±2.7 | 19±3.9 | 4.8±1.2 |
| Guiyang | 30 | 227±77 | 7.5±4.4 | 0.76±0.5 | 6.0±3.4 | 11±4.4 |
| Chengdu | 26 | 105±39 | 17.7±8.1 | 1.8±0.8 | 28±4.8 | 10±3.0 |
| Wuhan | 22 | 123±49 | 17.5±8.3 | 2.0±1.2 | 24±8.5 | 9.6±2.7 |
| Guangzhou | 28 | 85±32 | 17.4±9.9 | 2.3±1.8 | 33±11 | 8.1±2.4 |
| Nanjing | 19 | 111±50 | 18.8±8.7 | 1.6±0.6 | 28±9.3 | 12±3.8 |
| Shanghai | 27 | 68±43 | 7.2±9.0 | 1.0±0.9 | 17±8.5 | 7.4±3.0 |





Table 2 Proportion of modern carbon in WSOC, WINSOC, OC, EC, TC, and anhydrosugar, and ratio data for 10 urban cites in China for the period October 2013 to November 2013

| | Start date | PM$_{2.5}$ | WSOC | WINSOC | EC | f$_{m(WSOC)}$ | f$_{m(WINSOC)}$ | f$_{m(OC)}$ | f$_{m(EC)}$ | f$_{m(TC)}$ | Lev | Lev/OC | Gal | Man |
|---|---|---|---|---|---|---|---|---|---|---|---|---|---|---|
| BJ1 | 11/3/2013 | 88 | 5.49 | 5.62 | 1.4 | 0.72 | 0.73 | 0.72 | 0.51 | 0.70 | 176 | 15.9 | 31.7 | 65.1 |
| BJ2 | 11/5/2013 | 298 | 23.7 | 29.2 | 6.47 | 0.63 | 0.67 | 0.65 | 0.49 | 0.63 | 398 | 7.50 | 38.6 | 79.3 |
| XX1 | 10/15/2013 | 132 | 4,71 | 17.7 | 4.30 | 0.65 | 0.51 | 0.54 | 0.38 | 0.51 | 553 | 24.7 | 29.3 | 52.1 |
| XX2 | 10/22/2013 | 320 | 9.29 | 39.8 | 6.73 | 0.64 | 0.63 | 0.63 | 0.35 | 0.60 | 601 | 12.3 | 31.8 | 60.8 |
| TY1 | 10/25/2013 | 177 | 15.9 | 12.5 | 5.90 | 0.81 | 0.66 | 0.74 | 0.44 | 0.69 | 518 | 18.2 | 28.4 | 56.4 |
| TY2 | 10/26.2013 | 314 | 26.9 | 26.9 | 14.2 | 0.58 | 0.52 | 0.55 | 0.28 | 0.50 | 672 | 12.5 | 36.3 | 86.4 |
| LZ1 | 10/20/2013 | 123 | 13.8 | 2.81 | 3.74 | 0.72 | 0.58 | 0.70 | 0.56 | 0.67 | 442 | 26.7 | 22.6 | 53.8 |
| LZ2 | 10/23/2013 | 199 | 25.1 | 7.64 | 7.51 | 0.67 | 0.65 | 0.66 | 0.42 | 0.62 | 439 | 13.4 | 21.4 | 51.5 |
| GY1 | 10/31/2013 | 125 | 3.74 | 1.18 | 0.64 | 0.57 | 0.81 | 0.63 | 0.71 | 0.64 | 247 | 50.1 | 16.4 | 35.5 |
| GY2 | 11/6/2013 | 287 | 9.41 | 4.36 | 2.04 | 0.52 | 0.78 | 0.61 | 0.55 | 0.60 | 436 | 31.7 | 24.7 | 64.6 |
| CD1 | 10/31/2013 | 53.8 | 4.40 | 0.86 | 0.63 | 0.87 | 0.55 | 0.82 | 0.51 | 0.79 | 198 | 37.6 | 13.2 | 21.2 |
| CD2 | 11/8/2013 | 109 | 14.7 | 5.59 | 4.77 | 0.78 | 0.71 | 0.76 | 0.49 | 0.71 | 368 | 18.2 | 27.9 | 46.6 |
| WH1 | 10/26/2013 | 73.2 | 13.0 | 3.59 | 1.40 | 0.69 | 0.71 | 0.69 | 0.42 | 0.67 | 344 | 20.7 | 15.1 | 32.0 |
| WH2 | 10/30/2013 | 182 | 25.9 | 18.1 | 4.94 | 0.75 | 0.73 | 0.74 | 0.54 | 0.72 | 324 | 7.37 | 16.3 | 30.1 |
| NJ1 | 10/27/2013 | 88.2 | 14.3 | 2.04 | 1.48 | 0.73 | 0.62 | 0.72 | 0.51 | 0.70 | 235 | 14.4 | 11.9 | 23.7 |
| NJ2 | 10/29/2013 | 149 | 26.5 | 7.91 | 3.42 | 0.65 | 0.63 | 0.64 | 0.43 | 0.63 | 520 | 15.1 | 18.6 | 30.9 |
| GZ1 | 10/28/2013 | 67.2 | 7.40 | 3.89 | 2.20 | 0.79 | 0.64 | 0.74 | 0.41 | 0.68 | 161 | 14.3 | 10.5 | 25.3 |
| GZ2 | 10/29/2013 | 149 | 23.1 | 20.7 | 5.55 | 0.69 | 0.58 | 0.64 | 0.24 | 0.59 | 279 | 6.37 | 13.7 | 35.6 |
| SH1 | 10/20/2013 | 63.2 | 6.39 | 1.70 | 1.58 | 0.78 | 0.57 | 0.73 | 0.56 | 0.71 | 165 | 20.4 | 9.77 | 19.7 |
| SH2 | 10/23/2013 | 209 | 23.8 | 18.2 | 5.72 | 0.75 | 0.60 | 0.68 | 0.33 | 0.67 | 468 | 11.1 | 18.8 | 37.2 |

Note: all fractions are in μg m$^{-3}$, except for levoglucosan (Lev), galactosan (Gal) and mannosan (Man) (all ng m$^{-3}$).





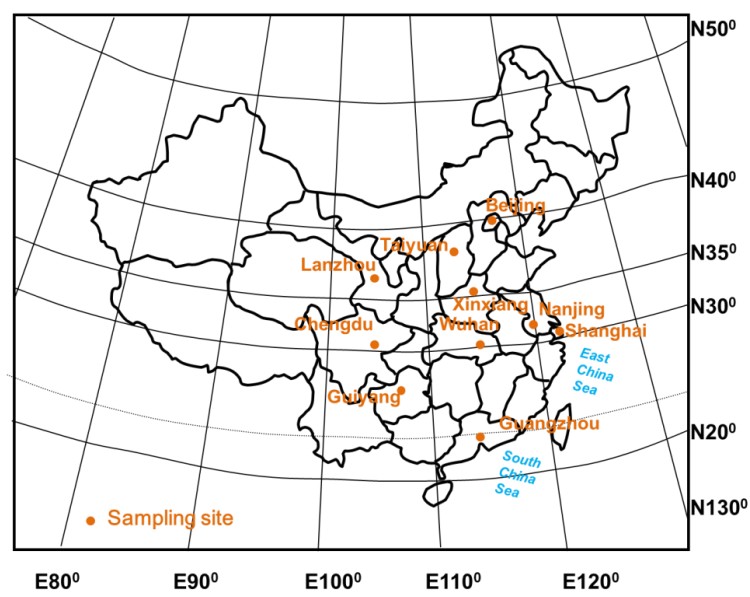


Figure 1. Geographic locations of the 10 Chinese sampling sites.














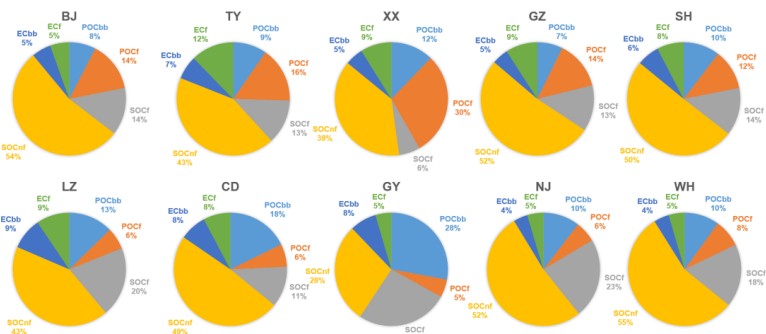


Figure 2. The proportions of different carbon fractions, including elemental carbon derived from
fossil fuels (1EC$_f$), EC derived from burning biomass (EC$_{bb}$), BB-derived primary organic carbon
(POC$_{bb}$), POC derived from FF (POC$_f$), non-FF secondary OC (SOC$_{nf}$) and SOC derived from FF
(SOC$_f$) in total carbon (TC) for 10 urban cites during the sampling period.





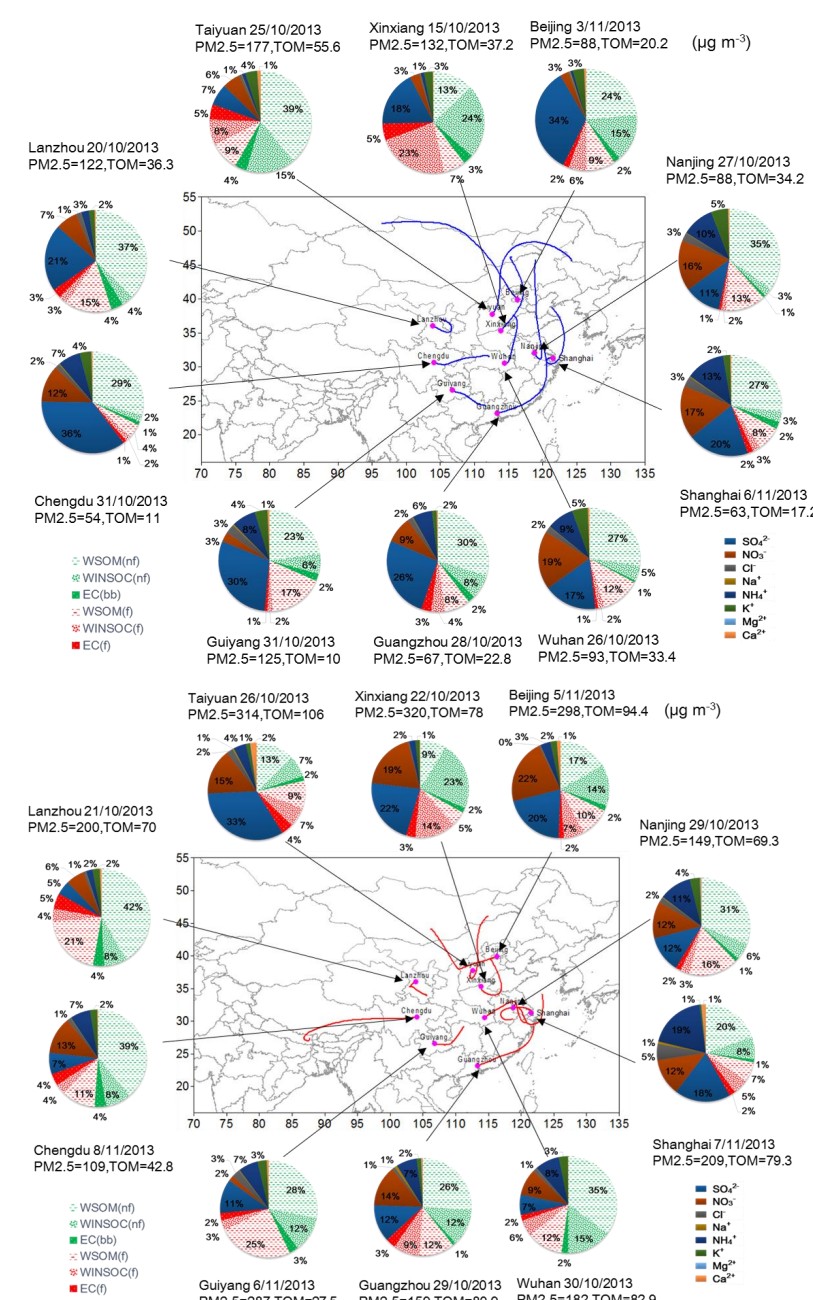



Figure 3. The chemical compositions of fine particles (PM$_{2.5}$) under non-haze (top) and haze

(bottom) conditions during the sampling period.




The English in this document has been checked by at least two professional editors,
both native speakers of English. For a certificate, please see:

http://www.textcheck.com/certificate/foV8Qb