# Peer review of "Sources of non-fossil fuel emissions in carbonaceous aerosols during early winter in Chinese"

_Atmospheric Chemistry and Physics, 2017_

## Referee Comment (RC1) · Anonymous Referee #1 · 8 Jun 2017

This study attempts to elucidate sources of OC (WSOC, WINSOC) and EC using 14C and molecular organic tracers. Such methods have already been successfully applied in many regions around the world. This study found non-fossil fuel (NF) emissions were predominant in total carbon. Primary organic carbon was very important in North China. Given that the powerful property of radiocarbon in determining the sources of fossil and nonfossil and board implications present in this study, I recommend it for a publication in ACP after some revisions required below.

Source apportionment of POC (NF+FF) and SOC (NF+FF) is based on several assumptions, which should be carefully evaluated and clearly indicated in the paper. If POC and SOC numbers are shown in the abstract and conclusions, the authors should also point out assumptions and limitations in POC and SOC estimations in the abstract

[Figure]

and conclusions as comments provided below.

Specifics

Line 39: "dominant" is too strong to be used here. Line 46-49: to include coal combustion in fossil fuel emissions. Line 51: to add references? Lines 52-56: the sentence should be reworded. a large fraction of SOA can be water insoluble as well. Line 72-74: references related to recent studies in China should be included here.

Method part: sample numbers for all measurements should be clearly shown in the text and tables/Figure captions.

Line 107-108: more details should be provided. Line 106: What are the uncertainties of $f_{NF}$ (and $f_M$) in WSOC, WINSOC, OC and EC? Lines 138-140: should be removed because no evidence was provided. Lines 150-170: please compare your data with published results (e.g. Beijing)? Why the biomass burning contribution to EC in Beijing was ~50%, which was much higher than those from other studies in the same city? Since only 2 samples were selected for each city, did these two samples can represent the winter? I suggest limitations should be pointed out clearly. 171-175: to add comparisons with published results in China and also other sites in Asia. Line 203: why 7.7 6 $\pm$ 1.47 $((OC/Lev)_{bb})$? This can be estimated by ïïjŽ $OC_{BB}=(OC/EC)_{BB}*EC_{BB}$ as well. Line 206: $SOC_{nf} = OC_{nf} - POC_{bb}$ is not correct. Non-fossil source should at least include BB, SOC as well biogenic emissions and cooking, Line 208: $POC_f =$ WINSOC $\times$ $(1-f_c)$ is not correct. A large fraction WINSOC can be from secondary organic aerosol as well. So $POC_f$ is an upper limit of POCF. This should be carefully pointed out and discussed. And please add references after Eq 3-6. Lines 236-246: Please discuss the possible biased in SOC estimations based on Eq 3-6. Line 227-230: How do you exclude contribution from residential coal combustion? I suggest removing the discussion if no other evidence can be found.

---

## Referee Comment (RC2) · Anonymous Referee #2 · 9 Jun 2017

This manuscript applied a powerful radiocarbon source tracer to apportion fossil fuel and biomass/biofuel contributions to carbonaceous aerosols in ten cities of China. The method was well established. Although the sample numbers are limited for each city (two samples), the result contain new message for sources of organic carbon, elemental carbon, water soluble organic carbon, primary and secondary aerosols in Chinese cities. These carbonaceous aerosols are included as major concerns for climate changes and human health. The conclusion therefore is important for air pollution mitigation in China. Before publication on ACP, some technical improvements are suggested.

Line 46, "fossil fuels " changes to fossil fuel combustion.

Line 56, 2007b;Docherty et al., 2008;Mayol‐Bracero et al., 2002;Weber et al.,

2007a); (Huang et al., 2014). Error.

Line 57, Several methods have been introduced to identify and quantify OC emission sources. Please show more methods for aerosol source apportionment; other methods like receptor models (PMF, CMB), and dispersion models.

Line 65 14C level (Szidat et al., 2009) Hence, 14C measurements can provide information about the. Full stop had been omitted.

Line 66: Numerous studies have been performed on the regional background of carbonaceous aerosols at urban sites. I prefer to change this sentence to: Numerous studies have been performed at urban sites to assess carbonaceous aerosol sources at the regional scale.

Line 68: contemporary carbon was the dominant pollutant in carbonaceous aerosols at a background site; The references should be cited for this conclusion at a background site (which one, it is better to detail the background site).

while a significant difference was found among seasons at urban sites (Yang et al., 2005;Chen et al., 2013;Liu et al., 2013a;Zhang et al., 2014b;Liu et al., 2014a). This is a new/independent sentence which suggests seasonal variations at urban sites. The conjunction word "while" is not suitable since the seasonal variations have no clear relationship with the previous result from a background site.

Line 72: aerosols (Gelencsér et al., 2007;Ding et al., 72 2008;Lee et al., 2010;Yttri et al., 2011). It is better to add one or two latest references. The combination of organic tracer and radiocarbon diagnosing is the main advantage of this research. Therefore, it should have one or two latest literatures to support the hot topic of this method.

Line 74: the beginning of the period of widespread hazes. Where? Probably it may be specified in China.

Line 75: carbon fractions such as WSOC, WINSOC and EC, along with water-soluble inorganic ions (F−, Cl−, SO42-, NO3-, NH4 , Na+, K+, Ca2+ and Mg2+) and anhydrosugars (levoglucosan, galactosan and mannosan). The details of water-soluble inorganic ions and anhydrosugars in brackets should not be showed in the introduction, while they should appear in method or result.

The last paragraph of Introduction, authors may include some information for the advantage of the combination of radiocarbon and anhydrosugar tracer. In introduction, authors should clarify what are target sources for organic tracer.

Fig.1, I suggest to include annual or winter aerosol optical depth to display the representative of the 10 cities for air pollution hotspots in China. Alternative, a literature for PM2.5 map in China may be helpful to show the relative high levels of the 10 cities. An example can be found in figure 1 of a publication: Light absorption enhancement of black carbon from urban haze in Northern China winter, Environ. Pollut., 221, 418-426, doi: http://dx.doi.org/10.1016/j.envpol.2016.12.004.

I am interesting on the thermal and FID signal of the EC isolation of radiocarbon analysis of this method. This method is similar to CTO-375, but different from SWISS-4 (i.e. Zhang et al.) and NIOSH870 protocols.

Line 308: PM2.5, OC and EC levels were highest in northern China, with maximum concentrations of 482 $\mu$g m-3, 75.9 $\mu$g m-3 and 19.3 $\mu$g m-3, respectively. Please show the detail site of these highest levels.

Line 309: OC and EC were the major components of PM2.5, accounting for 13 $\pm$ 8% and 2 $\pm$ 1%, of total PM2.5, respectively. This is not suitable conclusion of this study. Author did not analyze several major chemicals such as sulfate, nitrate. I do agree that OC and EC are very important species of particulate matter, considering the health and climate impacts of the carbonaceous aerosols.

Line 320: while SOC contributed more in cities in other regions of China. What is the meaning of other regions in China? Please specify the exact regions.

Line 321-322: however, the contribution of POC from both NF and NF increased significantly in these periods. This sentence should be corrected and improved.

Final sentence: This indicates that synoptic conditions promote the accumulation of particles derived either from local or regional sources. This is not an informative conclusion for the scope of this research.

---

## Author Comment (AC1) · 16 Jul 2017

**Response to Reviewers' Comments and Suggestions**

**Comments from Referees**
*Author's response*
Author's changes in manuscript

**Comments:**

**This study attempts to elucidate sources of OC (WSOC, WINSOC) and EC using 14C and molecular organic tracers. Such methods have already been successfully applied in many regions around the world. This study found non-fossil fuel (NF) emissions were predominant in total carbon. Primary organic carbon was very important in North China. Given that the powerful property of radiocarbon in determining the sources of fossil and nonfossil and board implications present in this study, I recommend it for a publication in ACP after some revisions required below.**

**Source apportionment of POC (NF+FF) and SOC (NF+FF) is based on several assumptions, which should be carefully evaluated and clearly indicated in the paper. If POC and SOC numbers are shown in the abstract and conclusions, the authors should also point out assumptions and limitations in POC and SOC estimations in the abstract and conclusions as comments provided below.**

**Line 39: "dominant" is too strong to be used here.**
*Response and Revisions*:*Thank you for your suggestion. The "dominant" has been revised into "important."*
Author's changes in manuscript: "Carbonaceous aerosols are the important component of PM2.5 (~20–80%)."

**Line 46-49: to include coal combustion in fossil fuel emissions.**
*Response and Revisions*:*The "coal combustion" has been added into fossil fuel emissions.*
Author's changes in manuscript: "vehicle or industry emissions such as coal combustion"

**Line 51: to add references?**
*Response and Revisions*:*The reference has been added.*
Author's changes in manuscript: "formed through oxidation of reactive organic gases followed by gas-to-particle conversion in the atmosphere (secondary OC; SOC) (Choi et al., 2012;Subramanian et al., 2007)"

**Lines 52-56: the sentence should be reworded. a large fraction of SOA can be water insoluble as well.**
*Response and Revisions*:*Thank you for pointing out this. We have already changed "WINSOC better represents POC" into "a large fraction of WINSOC is from POC" (line 56)*
Author's changes in manuscript: "while a large fraction of WINSOC is from POC"

**Line 72-74: references related to recent studies in China should be included here.**

*Response and Revisions*:*The references regarding recent studies in China have been added in the revised manuscript.*

Author's changes in manuscript: "A combination of 14C analysis and organic tracer determination allows for more detailed source apportionment of carbonaceous aerosols (Gelencsér et al., 2007;Ding et al., 2008;Lee et al., 2010;Yttri et al., 2011;Zong et al., 2016;Liu et al., 2015;Zhang et al., 2014b)"

**Method part: sample numbers for all measurements should be clearly shown in the text and tables/Figure captions.**

*Response and Revisions*:*Thank you for your suggestion. The sample numbers for all measurements have already shown in the text and tables/Figure captions.*

Author's changes in manuscript: "All samples were analyzed for OC and EC, and 20 samples, including two filters based on the PM2.5 concentrations at each site, were selected for further chemical analysis."

**Line 107-108: more details should be provided.**

*Response and Revisions*:*More details have already been added into the revised manuscript (line 123-138). With regard to more detailed method development of 14C analysis of WINSOC and EC please see at http://pubs.acs.org/doi/abs/10.1021/es401250k?journalCode=esthag (Title: The use of levoglucosan and radiocarbon for source apportionment of PM2.5 carbonaceous aerosols at a background site in East China). In addition, detailed information of 14C analysis of WSOC, WINSOC and EC can be found at* http://pubs.acs.org/doi/abs/10.1021/es503102w *(Title: Source Apportionment Using Radiocarbon and Organic Tracers for PM2.5 Carbonaceous Aerosols in Guangzhou, South China: Contrasting Local- and Regional-Scale Haze Events).*

Author's changes in manuscript: "To obtain the WSOC, WINSOC and EC fractions from a single punch filter, a circular section of the punch filter was clamped in place between a filter support and a funnel and then 60 ml ultra-pure water was slowly passed through the punch filter without a pump, allowing the WSOC to be extracted delicately. WSOC was quantified as the total dissolved organic carbon in solution using a total organic carbon (TOC) analyzer (Shimadzu TOC_VCPH, Japan) following the nonpurgeable organic carbon protocol. WSOC solution was freeze-dried to dryness at -40 °C. The WSOC residue was re-dissolved with ~500 µl of ultra-pure water and then transferred to a pre-combusted quartz tube, which was then placed in the freeze dryer. After that, the quartz tube was combusted at 850 °C. The remaining carbon on the filter was identified as WINSOC or EC by an OC/EC analyzer (Sunset, U.S.). After WSOC pretreatment and freeze-dried, OC is oxidized to $CO_2$ under a stream of pre-cleaned oxygen pure analytical grade $O_2$ (99.999%, 30 ml min-1) during the pre-combustion step at 340°C for 15 min. Before the OC is oxidized, the sample is first positioned in the 650 °C oven for about 45 s flash heating. This flash heating has the advantage of minimizing pre-combustion charring, since it reduces pyrolysis of OC. After the OC separation, the filters were removed from the system, placed into a muffle furnace at 375°C, and combusted for 4 h. The filters were then quickly introduced back into the

system and oxidized under a stream of pure oxygen at 650°C for 10 min to analyze the EC fraction."

**Line 106: What are the uncertainties of fNF (and fM) in WSOC, WINSOC, OC and EC?**

*Response and Revisions:The uncertainties of fNF (and fM) in WSOC, WINSOC, OC and EC were up to 20%,20%,15% and 15%, respectively. (line 143-144)*

Author's changes in manuscript: "The uncertainties of fnf (and fm) in WSOC, WINSOC, OC and EC were up to 20%,20%,15% and 15%, respectively."

**Lines 138-140: should be removed because no evidence was provided.**

*Response and Revisions:The sentence was removed.*

Author's changes in manuscript:

**Lines 150-170: please compare your data with published results (e.g. Beijing)? Why the biomass burning contribution to EC in Beijing was ∽50%, which was much higher than those from other studies in the same city? Since only 2 samples were selected for each city, did these two samples can represent the winter? I suggest limitations should be pointed out clearly.**

*Response and Revisions:Thank you for your comments. The reasons for higher biomass burning contribution to EC in Beijing maybe attributed to (1) **different method for isolation of OC and EC for 14C determination**. In this study, OC and EC separation was based on their different thermal behavior, which is different from other methods such as thermal-optical method. Our results were comparable with the same approach carried out in Beijing (~50%) (please see article at* http://pubs.acs.org/doi/abs/10.1021/es503102w*); and (2) **samples selection**. We only selected two filter samples based on relatively lower and higher PM2.5 concentration for each site. The reasons for sample selection are 1) to see difference between haze and non-haze episode during winter campaign, and 2) limitations for 14C analysis such as OC/EC separation technique, the bulk samples required, and the high cost for 14C measurement. These selection choices may influence the final results to some extent. The limitation have already been added in the revised manuscript (line 189-192).*

Author's changes in manuscript: "Compared with other studies in China, the measured biomass burning contributions to EC in Beijing are relatively higher than those in the same city during winter (Zhang et al., 2014b;Zhang et al., 2015b). This is due to the fact that different approach we used for OC/EC separation, and sample selection in this study (selected two filter samples based on relatively lower and higher PM2.5 concentration for each site) because of limitations for 14C analysis (i.e. the bulk samples required and the high cost for 14C measurement). However, the result is similar with those using the same approach (Liu et al., 2016c;Zong et al., 2016)."

171-175: to add comparisons with published results in China and also other sites in Asia.

*Response and Revisions:Thank you for your suggestion. The comparisons have already been added into the revised manuscript (line 187-190).*

Author's changes in manuscript: "Over half of the OC fraction was from NF sources at all sites (range: 54–82%), with an average NF source contribution of 68 ± 7%, comparable to previous study reported in four Chinese cities during 2013 winter (Xi'an,

Beijing, Shanghai and Guangzhou were 63%, 42%, 51% and 65%, respectively)(Zhang et al. 2015a)."

**Line 203: why 7.76 ± 1.47 ((OC/Lev) bb)? This can be estimated by :OCBB=(OC/EC)BB *ECBB as well.**

*Response and Revisions:Yes, but here we have measured a good tracer of biomass burning emissions so we used (OC/Lev) bb for the estimation.*
Author's changes in manuscript:

**Line 206: SOC nf = OC nf - POC bb is not correct. Non-fossil source should at least include BB, SOC as well biogenic emissions and cooking.**

*Response and Revisions:Thank you for your comments. SOCnf may also include other non-fossil sources such as cooking and biogenic emissions, however, they should be limited during wintertime (e.g., <20%). Therefore, our estimates of SOC many generally represent an upper limit but this will not change our conclusion towards to the spatial distribution of SOC in China.*
Author's changes in manuscript:

**Line 208: POCf = WINSOC × (1-fc) is not correct. A large fraction WINSOC can be from secondary organic aerosol as well. So POCf is an upper limit of POCF. This should be carefully pointed out and discussed. And please add references after Eq 3-6.**

*Response and Revisions:Thank you for your comments. Yes, this is an upper limit of POCf. This sentence was added in the revised manuscript (line 259-260). Related articles on this model have already been added (line: 227-228).*
Author's changes in manuscript: "Typical relative uncertainties were recently estimated, using a similar modelling approach, at 20–25 % for SOCnf, SOCf, POCbb, and POCf, and ~13% for ECf, and ECbb (Zhang et al., 2015a). A large fraction WINSOC can be from secondary organic aerosol as well. Hence POCf is an upper limit of POCf. SOCf may be overestimated if a small fraction (e.g. <20%) WSOC is not secondary, so SOCf may be an upper limit. Meanwhile, SOCnf  may also include other non-fossil sources such as cooking and biogenic emissions, however, they should be limited during wintertime (e.g., <20%). Therefore, our estimates of SOC many generally represent an upper limit but this will not change our conclusion towards to the spatial distribution of SOC in China."

**Lines 236-246: Please discuss the possible biased in SOC estimations based on Eq 3-6.**

*Response and Revisions:We have already added the sentences.*
Author's changes in manuscript: "SOCf may be overestimated if a small fraction (e.g. <20%) WSOC is not secondary, so SOCf may be an upper limit. Meanwhile, SOCnf may also include other non-fossil sources such as cooking and biogenic emissions, however, they should be limited during wintertime (e.g., <20%). Therefore, our estimates of SOC many generally represent an upper limit but this will not change our conclusion towards to the spatial distribution of SOC in China."

**Line 227-230: How do you exclude contribution from residential coal combustion? I suggest**

**removing the discussion if no other evidence can be found.**

*Response and Revisions*:*The sentence has already been revised.*

Author's changes in manuscript: "The ratios of POCf to ECf (0.66–3.32) were within the emission ratios between coal combustion (2.7–6.1) (Zhang et al., 2008) and traffic exhausts fumes (0.5–1.3) (Zhou et al., 2014;He et al., 2008), indicating that coal combustion and traffic exhaust fumes were the major primary sources at all sites."

---

## Author Comment (AC2) · 16 Jul 2017

**Response to Reviewers' Comments and Suggestions**

*Reviewer's Comments*
Authors' responses and revisions
**Comments from Referees**
*Author's response*
Author's changes in manuscript

**Reviewer: 2**

**Comments:**

**This manuscript applied a powerful radiocarbon source tracer to apportion fossil fuel and biomass/biofuel contributions to carbonaceous aerosols in ten cities of China. The method was well established. Although the sample numbers are limited for each city (two samples), the result contain new message for sources of organic carbon, elemental carbon, water soluble organic carbon, primary and secondary aerosols in Chinese cities. These carbonaceous aerosols are included as major concerns for climate changes and human health. The conclusion therefore is important for air pollution mitigation in China. Before publication on ACP, some technical improvements are suggested.**

**Line 46, "fossil fuels" changes to fossil fuel combustion.**
*Response and Revisions*:*"fossil fuels" has already changed into "fossil fuel combustion."*
Author's changes in manuscript: "EC is formed either from biomass burning (BB; e.g., wood fires, heating) or fossil fuel combustion"

**Line 56, 2007b;Docherty et al., 2008;MayolâA˘ RBracero et al., 2002;Weber et al., 2007a); (Huang et al., 2014). Error.**
*Response and Revisions*:*Thank you for pointing out this. We have already made the correction in the revised manuscript (line 56-57).*
Author's changes in manuscript: "(Weber et al., 2007b;Docherty et al., 2008;Mayol‐Bracero et al., 2002;Weber et al., 2007a;Huang et al., 2014)"

**Line 57, Several methods have been introduced to identify and quantify OC emission sources. Please show more methods for aerosol source apportionment; other methods like receptor models (PMF, CMB), and dispersion models.**
*Response and Revisions*:*Thank you for your suggestion. The references have already added in the revised manuscript (line 59-60).*
Author's changes in manuscript: "Several methods have been introduced to identify and quantify OC emission sources, such as the use of organic molecular tracers (Simoneit et al., 1999), receptor models (PMF, CMB)(Singh et al., 2017;Bove et al., 2014;Marcazzan et al., 2003), and dispersion models (Colvile et al., 2003);"

**Line 65 14C level (Szidat et al., 2009) Hence, 14C measurements can provide information about the. Full stop had been omitted.**

*Response and Revisions*:*Thank you for pointing out this. Full stop has already added in the revised manuscript.*

Author's changes in manuscript: "shows a high contemporary 14C level (Szidat et al., 2009)."

**Line 66: Numerous studies have been performed on the regional background of carbonaceous aerosols at urban sites. I prefer to change this sentence to: Numerous studies have been performed at urban sites to assess carbonaceous aerosol sources at the regional scale.**

*Response and Revisions*:*This sentence has been changed.*

Author's changes in manuscript: "Numerous studies have been performed at urban sites and background sites to assess carbonaceous aerosol sources."

**Line 68: contemporary carbon was the dominant pollutant in carbonaceous aerosols at a background site; The references should be cited for this conclusion at a background site (which one, it is better to detail the background site).**

*Response and Revisions*:*We have already added references and pointed out the detailed background sites in the revised manuscript (line 70-73).*

Author's changes in manuscript: "For example, contemporary carbon was the dominant pollutant in carbonaceous aerosols at a background sites such as Ningbo and Hainan stations (Liu et al., 2013a;Zhang et al., 2014c)"

**while a significant difference was found among seasons at urban sites (Yang et al., 2005;Chen et al., 2013;Liu et al., 2013a;Zhang et al., 2014b;Liu et al., 2014a). This is a new/independent sentence which suggests seasonal variations at urban sites. The conjunction word "while" is not suitable since the seasonal variations have no clear relationship with the previous result from a background site.**

*Response and Revisions*:*We are sorry for the misunderstanding and thank you for your suggestion. We have already revised the sentence.*

Author's changes in manuscript: "In urban, the relative carbon contributions have shown a significant seasonal difference (Yang et al., 2005;Chen et al., 2013;Liu et al., 2013b;Zhang et al., 2014a;Liu et al., 2014a;Zhang et al., 2017)"

**Line 72: aerosols (Gelencsér et al., 2007;Ding et al., 72 2008;Lee et al., 2010;Yttri et al., 2011). It is better to add one or two latest references. The combination of organic tracer and radiocarbon diagnosing is the main advantage of this research. Therefore, it should have one or two latest literatures to support the hot topic of this method.**

*Response and Revisions*:*Thank you for pointing out this. We have already added new literatures in the revised manuscript (line 76-77).*

Author's changes in manuscript: "A combination of 14C analysis and organic tracer determination allows for more detailed source apportionment of carbonaceous aerosols (Gelencsér et al., 2007;Ding et al., 2008;Lee et al., 2010;Yttri et al., 2011;Zong et al., 2016;Liu et al., 2015;Zhang et al., 2014b)"

**Line 74: the beginning of the period of widespread hazes. Where? Probably it may be specified**

**in China.**

*Response and Revisions: We are sorry for the misunderstanding and thank you for your suggestion. We have revised the sentence.*

Author's changes in manuscript: "In this study, sampling was conducted in 10 typical Chinese cities during early winter when heavy haze pollution frequently occurs in this season."(line 78-79)

**Line 75: carbon fractions such as WSOC, WINSOC and EC, along with water-soluble inorganic ions (F-, Cl-, SO42-, NO3-, NH4 , Na+, K+, Ca2+ and Mg2+) and anhydrosugars (levoglucosan, galactosan and mannosan). The details of water-soluble inorganic ions and anhydrosugars in brackets should not be showed in the introduction, while they should appear in method or result.**

*Response and Revisions:Thank you for your suggestion. We have revised the sentence.*

Author's changes in manuscript: "Carbonaceous aerosols, including different carbon fractions such as WSOC, WINSOC and EC, along with water-soluble inorganic ions and anhydrosugars, were analyzed in PM2.5 samples." The details of compounds have already shown in the method section.

**The last paragraph of Introduction, authors may include some information for the advantage of the combination of radiocarbon and anhydrosugar tracer. In introduction, authors should clarify what are target sources for organic tracer.**

*Response and Revisions:Thank you for your suggestion. We have already added the sentence as following.*

Author's changes in manuscript: "In particular, anhydrosugars such as levoglucosan are used as a molecular marker to indicate biomass-burning emissions. The combination of 14C analysis and the concentration of levoglucosan has offered new insights into the detailed sources of carbonaceous aerosols. So, source apportionment of carbonaceous aerosols was performed using a source apportionment model based on the 14C results and measured chemicals."

**Fig.1, I suggest to include annual or winter aerosol optical depth to display the representative of the 10 cities for air pollution hotspots in China. Alternative, a literature for PM2.5 map in China may be helpful to show the relative high levels of the 10 cities. An example can be found in figure 1 of a publication: Light absorption enhancement of black carbon from urban haze in Northern China winter, Environ. Pollut., 221, 418-426, doi: http://dx.doi.org/10.1016/j.envpol.2016.12.004.**

*Response and Revisions:Thank you for your suggestion. We have already added new figure 1 into the revised manuscript.*

**I am interesting on the thermal and FID signal of the EC isolation of radiocarbon analysis of this method. This method is similar to CTO-375, but different from SWISS-4 (i.e. Zhang et al.) and NIOSH870 protocols.**

*Response and Revisions:These methods utilize the difference in thermal stability between OC and EC, which is different from the method of SWISS-S using thermal-optical approach. C14 signal in*

*the EC fraction in this method was performed by evaporation of OC in a muffle furnace at 375°C in air with reaction time of 4h. More detailed method development of 14C analysis of WINSOC and EC please see at http://pubs.acs.org/doi/abs/10.1021/es401250k?journalCode=esthag (Title: The use of levoglucosan and radiocarbon for source apportionment of PM2.5 carbonaceous aerosols at a background site in East China). In addition, detailed information of 14C analysis of WSOC, WINSOC and EC can be found at* [http://pubs.acs.org/doi/abs/10.1021/es503102w](http://pubs.acs.org/doi/abs/10.1021/es503102w) *(Title: Source Apportionment Using Radiocarbon and Organic Tracers for PM2.5 Carbonaceous Aerosols in Guangzhou, South China: Contrasting Local- and Regional-Scale Haze Events).*

**Line 308: PM2.5, OC and EC levels were highest in northern China, with maximum concentrations of 482 μg m-3, 75.9 μg m-3 and 19.3 μg m-3, respectively. Please show the detail site of these highest levels.**

*Response and Revisions*:*We have already added details in the revised manuscript. (line 329)*
Author's changes in manuscript: "PM2.5 samples were collected continuously from 10 Chinese urban cities during early winter 2013. PM2.5, OC and EC levels were highest in northern China, with maximum concentrations of 482 μg m-3(Taiyuan, n=31), 75.9 μg m-3(Taiyuan, n=31) and 19.3 μg m-3(Beijing, n=31), respectively."

**Line 309: OC and EC were the major components of PM2.5, accounting for 13 ± 8% and 2 ± 1%, of total PM2.5, respectively. This is not suitable conclusion of this study. Author did not analyze several major chemicals such as sulfate, nitrate. I do agree that OC and EC are very important species of particulate matter, considering the health and climate impacts of the carbonaceous aerosols.**

*Response and Revisions*:*Thank you for pointing out this. The sentence has already been deleted.*

**Line 320: while SOC contributed more in cities in other regions of China. What is the meaning of other regions in China? Please specify the exact regions.**

*Response and Revisions*:*The sentence has already been revised into "while SOC contributed more in cities in other regions of China, such as Nanjing and Wuhan."*

**Line 321-322: however, the contribution of POC from both NF and NF increased significantly in these periods. This sentence should be corrected and improved.**

*Response and Revisions*:*This sentence has been changed into "During haze days, there were no dramatic changes in carbon sources or carbon compositions in the sampled cities, but the contributions of POC were relatively higher than the non-haze days."*

**Final sentence: This indicates that synoptic conditions promote the accumulation of particles derived either from local or regional sources. This is not an informative conclusion for the scope of this research.**

*Response and Revisions*:*Thank you for your suggestion. We have deleted the sentence.*